# MASKED INVERSE FOLDING WITH SEQUENCE TRANSFER FOR PROTEIN REPRESENTATION LEARNING

## ABSTRACT

Self-supervised pretraining on protein sequences has led to state-of-the art performance on protein function and fitness prediction. However, sequence-only methods ignore the rich information contained in experimental and predicted protein structures. Meanwhile, inverse folding methods reconstruct a protein's amino-acid sequence given its structure, but do not take advantage of sequences that do not have known structures. In this study, we train a masked inverse folding protein language model parameterized as a structured graph neural network. We then show that using the outputs from a pretrained sequence-only protein masked language model as input to the inverse folding model further improves pretraining perplexity. We evaluate both of these models on downstream protein engineering tasks and analyze the effect of using information from experimental or predicted structures on performance.

## 1 INTRODUCTION

Large pretrained protein language models (MLMs) have advanced the ability of machine-learning methods to predict protein structure, function, and fitness from sequence, especially when labeled training data is sparse. The recent state-of-the-art, inspired by BERT ((bidirectional encoder representations from transformers) (Devlin et al., 2018), uses increasingly-large transformer (Vaswani et al., 2017) models to reconstruct masked and mutated protein sequences taken from databases such as UniProt (UniProt Consortium, 2021), UniRef (Suzek et al., 2015), and BFD (Steinegger et al., 2019; Steinegger & Söding, 2018). Pretrained protein MLMs contain structural information (Rao et al., 2019; Rives et al., 2021; Chowdhury et al., 2021), encode evolutionary trajectories (Hie et al., 2022b; 2021), are zero-shot predictors of mutation fitness effects (Meier et al., 2021), improve out-of-domain generalization on protein engineering datasets (Dallago et al., 2021), and suggest improved sequences for engineering (Hie et al., 2022a). Protein MLMs are now incorporated into the latest machine-learning methods for detecting signal peptides (Teufel et al., 2021) and predicting intracellular localization(Thumuluri et al., 2022). However, only training on sequences ignores the rich information contained in experimental and predicted protein structures, especially as the number of high-quality structures from AlphaFold (Jumper et al., 2021; Varadi et al., 2022) increases.

Meanwhile, inverse folding methods reconstruct a protein's amino-acid sequence given its structure. Deep learning-based inverse folding is usually parametrized as a graph neural network (GNN) (Ingraham et al., 2019; Strokach et al., 2020; Jin et al., 2021; Jing et al., 2020) or SE(3)-equivariant transformer (McPartlon et al., 2022) that either reconstructs or autoregressively decodes the amino-acid sequence conditioned on the desired backbone structure. The ability to generate amino-acid sequences that fold into a desired structure is useful for developing novel therapeutics (Chevalier et al., 2017), biosensors (Quijano-Rubio et al., 2021), industrial enzymes (Siegel et al., 2010), and targeted small molecules (Lucas & Kortemme, 2020). Furthermore, single-chain inverse folding approaches could be coupled with recent sequential assembly based multimer structure prediction techniques (Bryant et al., 2022) for fixed-backbone multimer design.

However, we are primarily interested in using inverse folding as a pretraining task, with the intuition that incorporating structural information should improve performance on downstream tasks. Furthermore, current inverse folding methods must be trained on sequences with known or predicted structures, and thus do not take maximal advantage of the large amount of sequences that do not have known structures or of the menagerie of pretrained protein MLMs. For example, UniRef50

contains 42 million sequences, while the PDB (Rose et al., 2016) currently contains 190 thousand experimentally-measured protein structures.

In this study, we train a **M**asked **I**nverse **F**olding (MIF) protein masked language model (MLM) parameterized as a structured graph neural network (SGNN) (Ingraham et al., 2019). To our knowledge, this is the first example of combining the MLM task with structure in a pretraining task. We then show that using the outputs from a pretrained sequence-only protein MLM as input to MIF further improves pretraining perplexity by leveraging information from sequences without experimental structures. We will refer to this model as **M**asked **I**nverse **F**olding with **S**equence **T**ransfer (MIF-ST). Figure 1 compares the previous sequence-only dilated convolutional protein MLM (CARP), MIF, and MIF-ST. This is a novel way of transferring information from unlabeled protein sequences into a model that requires structure. We evaluate MIF and MIF-ST on downstream protein engineering tasks and analyze the effect of experimental and predicted structures on performance. Finally, we comment on the state of pretrained models for protein fitness prediction.

## 2 MIF AND MIF-ST

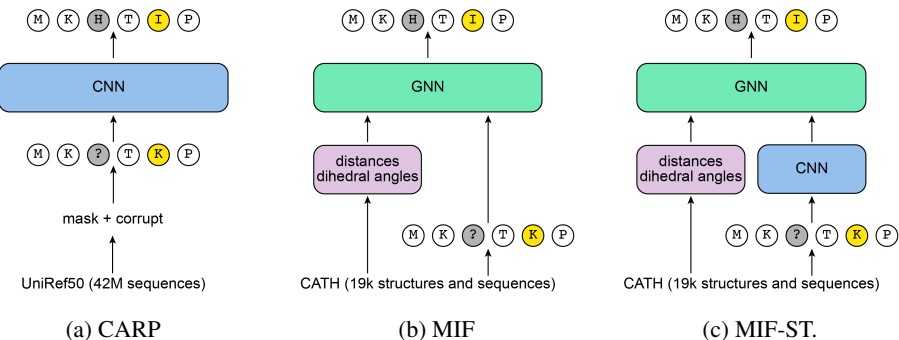

(a) CARP        (b) MIF        (c) MIF-ST.

Figure 1: Summary of models: (a) the **C**onvolutional **A**utoencoding **R**epresentations of **P**roteins protein masked language model, (b) the **M**asked **I**nverse **F**olding model, and (c) the **M**asked **I**nverse **F**olding with **S**equence **T**ransfer model.

### 2.1 BACKGROUND

Proteins are chains of amino acids that fold into three-dimensional structures. In masked language modeling pretraining on protein sequences, a model learns to reconstruct the original protein sequence from a corrupted version, and then the model likelihoods are used to make zero-shot predictions or the pretrained weights are used as a starting point for training on a downstream task, such as structure or fitness prediction. For example, ESM (Rives et al., 2021) and CARP (Yang et al., 2022) use the corruption scheme first described in BERT (Devlin et al., 2018). With a vocabulary of $\mathbb{T}$ of amino acids, we start from an amino-acid sequence $s$ of length $L$ of amino acids $s_i \in \mathbb{T} : 1 \leq i \leq L$, 15% of positions $\mathbb{M}$ are selected uniformly at random. 80% of these are changed to a special mask token, 10% are randomly mutated to another amino acid, and the remaining 10% are unchanged to generate $s_{\text{noised}}$. The model learns to predict the original amino acids:

$$p\left(s_i | s_{\text{noised}}\right) \forall i \in \mathbb{M} \tag{1}$$

by minimizing the negative log likelihood at positions $i \in \mathbb{M}$.

### 2.2 MASKED INVERSE FOLDING

While MLM pretraining on protein sequences can encode structural and functional information, adding information about the protein's backbone structure improves sequence recovery. A protein's backbone structure consists of the coordinates for each amino-acid residue's C, $C_\alpha$, $C_\beta$, and N atoms, leaving out information about the side chains (which would trivially reveal each residue's amino-acid

identity). We call the pretraining task of reconstructing a corrupted protein sequence conditioned on its backbone structure **M**asked **I**nverse **F**olding, which is illustrated in Figure 1b. We use the BERT corruption scheme and train the model to reconstruct the original amino acids conditioned on the corrupted sequence and the backbone structure:

$$p\left(\boldsymbol{s}_i|\boldsymbol{s}_{\text{noised}}, \text{structure}\right) \forall i \in \mathbb{M} \tag{2}$$

After pretraining, a MIF model can be used to perform any downstream task that a sequence-only PMLM can, with the caveat that a structure must be provided. Intuitively, structure-conditioned pretraining and having access to structures for the downstream task should both improve performance. We now discuss masked inverse folding in detail.

### 2.2.1 EMBEDDING BACKBONE STRUCTURE AND SEQUENCE

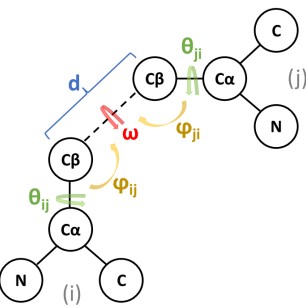

Figure 2: The backbone atoms amino-acid residues $i$ and $j$ with their dihedral and planar angles highlighted.

We represent protein backbone structures as graphs $\mathcal{G} = (\mathcal{V}, \mathcal{E})$ where each node $\mathcal{V}$ is an amino acid connected by edges $\mathcal{E}$ to its $k$-nearest amino-acid neighbors in the structure. We set $k = 30$ throughout. Each node's structural input features consist of the sine and cosine transformations of its dihedral and planar angles to its nearest neighbors in the primary structure:

$$\mathcal{V}_i = \{\sin, \cos\} \times \{\omega_{i,i+1}, \theta_{i+1,i}, \theta_{i,i+1}, \varphi_{i+1,i}, \varphi_{i,i+1}\} \in \mathbb{R}^{10} \tag{3}$$

Note that $\omega$ coordinates are symmetric whereas $\phi$ and $\psi$ coordinates are asymmetric and depend on residue order, so we encode both the forward and backwards angles in the forward and backward direction, i.e. $\phi_{i+1,i}$, and $\phi_{i,i+1}$, respectively. Figure 2 illustrates the backbone atoms of two residues and shows their dihedral and planar angles. Dihedral angles, planar angles, and residue distances used are unconventional to protein definitions and follow trRosetta (Yang et al., 2020) conventions.

The input edge features for the $i^{\text{th}}$ residue consist of the dihedral and planar angles and the Euclidean distance $d_{i,j \in N(i,k)}$ between the $C_\beta$ atom of residue $i$ and the $C_\beta$ atoms of its $k$-nearest neighbors, $N(i,k)$:

$$\mathcal{E}_{i,j} = \{d_{i,j}\} + \{\sin, \cos\} \times \{\omega_{i,j}, \theta_{i,j}, \theta_{j,i}, \varphi_{i,j}, \varphi_{j,i}\} \in \mathbb{R}^{k \times 11} \tag{4}$$

where $j \in N(i,k)$ and $j \neq i$.

We embed each token in $\boldsymbol{s}$ into a vector of size $d_s$:

$$\boldsymbol{S} = \text{embed}\left(\boldsymbol{s}\right) \in \mathbb{R}^{L \times d_s} \tag{5}$$

and concatenate it to the structural input node features to arrive at:

$$\boldsymbol{V} = \text{concat}\left(\mathcal{V}, \boldsymbol{S}\right) \in \mathbb{R}^{L \times (10 + d_s)} \tag{6}$$

Finally, we embed each edge and node into the model dimension $d$:

$$\boldsymbol{V}^0 = \boldsymbol{W}_v \boldsymbol{V} + \boldsymbol{b}_v \in \mathbb{R}^{L \times d} \qquad (7)$$

$$\boldsymbol{E}^0 = \boldsymbol{W}_e \mathcal{E} + \boldsymbol{b}_e \in \mathbb{R}^{L \times k \times d} \qquad (8)$$

Throughout, we set $d = 256$ and $d_s = 30$.

### 2.2.2 STRUCTURED GNN

We parametrize MIF as a bidirectional version of the structured GNN from Ingraham et al. (2019). The node and edge embeddings $\boldsymbol{V}^0$ and $\boldsymbol{E}^0$ are passed to a standard message-passing GNN with a multilayer perceptron aggregation function. The $m^{\text{th}}$ GNN layer takes as input node representations $\boldsymbol{V}^{m-1}$ and edge representations $\boldsymbol{E}^{m-1}$ and outputs $\boldsymbol{V}^m$ and $\boldsymbol{E}^m$. Within each layer, we first gather relational information from every neighboring node

$$R_i^m = \text{concat}\left( E_{j \in N(i,k)}^{m-1}, V_{j \in N(i,k)}^{m-1} \right) \in \mathbb{R}^{k \times 2d} \qquad (9)$$

and then compute the incoming messages at each node:

$$h_i^m = \text{Agg}\left[ f_{msg}(R_i^{(m)}) \right] \in \mathbb{R}^d \qquad (10)$$

We parameterize $f_{\text{msg}}$ as a three layer neural network with hidden dimension $d$ and ReLU non-linearities and Agg as a mean over the neighbor dimension. We then compute new node embeddings with another feed-forward neural network.

$$V_i^m = f_{\text{update}}\left( V_i^{m-1}, h_i^m \right) \qquad (11)$$

The sequence logits are computed as a linear mapping of the final node representations.

### 2.2.3 DATASETS AND MODEL TRAINING

We trained a 4-layer MIF on the CATH4.2 dataset Sillitoe et al. (2019) using the training, validation, and testing splits from Ingraham et al. (2019), in which there is no overlap between CATH topology classifications between data splits. MIF was trained with dynamic batch sizes to maximize GPU utilization with a maximum batch size of 6000 tokens or 100 sequences, the Adam optimizer, a maximum learning rate of 0.001, and a linear warmup over 1000 steps. Models were trained on one Nvidia V100 GPU for approximately one day, until validation perplexity stopped improving.

Previous work (Yang et al., 2022) trained CARP-640M, a dilated convolutional protein masked language model with approximately 640 million parameters trained on UniRef50 that achieves comparable results to the state-of-the-art transformer protein MLM, ESM-1b, which has a similar number of parameters and is trained on an earlier release of UniRef50. Importantly, all sequences with greater than 30% identity to the CATH test set were removed from CARP-640M's training set in order to obtain a fair evaluation on the CATH test set. As shown in Table 1, conditioning on the backbone structure drastically improves perplexity and sequence recovery compared to CARP-640M, despite MIF having 20 times fewer parameters and being trained on only the 19 thousand examples in CATH compared to the 42 million sequences in UniRef50. Increasing the GNN depth to 8 layers does not improve pretraining performance, so we use MIF with 4 layers for all following experiments. For comparison, we also train a 3.5M-parameter GVP (Jing et al., 2020) on the same masked inverse folding task (GVPMIF). The GVP architecture improves pretraining perplexity and recovery, but we find that it does not improve performance on downstream tasks.

### 2.3 MASKED INVERSE FOLDING WITH SEQUENCE TRANSFER

While conditioning on structure improves sequence recovery compared to sequence-only pretraining, we hypothesized that transferring information from sequences for which no structure is available should further improve performance on the pretraining task. Therefore, we transfer sequence

Table 1: Pretrained models. Parameters is the number of parameters trained on CATH4.2. Perplexity and recovery are on the CATH4.2 test set from (Ingraham et al., 2019).

| Regime | Model | Parameters | Perplexity | Recovery |
|---|---|---|---|---|
| sequence only | CARP-640M | 640M | 7.06 | 40.5% |
| sequence & structure | MIF-4 | 3.4M | 4.95 | 49.9% |
| | MIF-8 | 6.8M | 5.00 | 46.7% |
| | GVPMIF | 3.5M | 4.68 | 51.2% |
| +sequence transfer | MIF-ST | 3.4M | 4.08 | 55.6% |
| -UniRef50 pretraining | MIF-ST | 3.4M | 5.70 | 45.4% |

information from CARP-640M by directly replacing the sequence embedding in Equation 5 with the outputs from CARP-640M pretrained on UniRef50, as shown in Figure 1c. MIF-ST was trained with identical hyperparameters to MIF. The pretrained CARP-640M weights were not finetuned during training on CATH4.2. As shown in Table 1, sequence transfer improves perplexity and recovery on the CATH4.2 test set over both CARP-640M and MIF. Using the CARP-640M architecture with randomly-initialized weights as input to MIF did not improve performance, showing simply increasing model capacity is insufficient and that sequence transfer is necessary for the improvement.

## 3  RELATED WORK

**Sequence-only protein language models** A large body of work has recently studied the application of language models to sequence-only protein generation (Madani et al., 2020; 2021; Shin et al., 2021; Ferruz et al., 2022; Hesslow et al., 2022) and representation learning. See Wu et al. (2021) for a more comprehensive review of *de novo* protein sequence design with deep generative models. Alley et al. (2019) demonstrate that the internal representation learned by an LSTM-based autoregressive protein language model trained on a large protein sequence dataset can be leveraged for a wide variety of downstream tasks, including stability, fluorescence and secondary structure prediction. Rives et al. (2021); Rao et al. (2019; 2020); Elnaggar et al. (2021); Brandes et al. (2021) improve upon this by using the transformer architecture Vaswani et al. (2017), replacing the autoregressive pretraining task with a bidirectional denoising task, and scaling up the model and dataset sizes. Rao et al. (2021) further extend protein language models by allowing it to attend to multiple sequence alignments. (Yang et al., 2022) replace the transformer attention module with dilated convolutions. MIF and MIF-ST build on this body of work by combining the denoising task with structural conditioning.

**Fixed-backbone protein design** This is similar in spirit to work in fixed-backbone protein design, which involves the design of proteins with a given target backbone structure. Outside of deep-learning based methods, researchers use packing algorithms (Dahiyat & Mayo, 1997; Street & Mayo, 1999; DeGrado et al., 1991; Harbury et al., 1998), physics-based energy functions (Alford et al., 2017), or match structural motifs to sequence motifs (Zhou et al., 2020). More recent methods attempt to invert deep-learning protein structure prediction models (Jendrusch et al., 2021; Moffat et al., 2021; 2022; Anishchenko et al., 2021; Norn et al., 2021; Wang et al., 2021). See Ovchinnikov & Huang (2021) for a more comprehensive review of fixed-backbone protein design approaches.

Our method is most similar to work that directly conditions sequence generation on an encoding of the backbone structure. Ingraham et al. (2019) conditions an autoregressive sequence model on inter-residue distances and angles. Although Ingraham et al. (2019) focuses on a Structured Transformer, they note that replacing the transformer attention mechanism with a simple multilayer perceptron improves performance, and we use this Structured GNN architecture in both MIF and MIF-ST. Jing et al. (2020) improves on the results in Ingraham et al. (2019) by modeling all the backbone coordinates with a new geometric vector perceptron (GVP) architecture. McPartlon et al. (2022) replaces GVP with an SE(3)-equivariant transformer. Dauparas et al. (2022) and Shi et al. (2022) further improve performance on the inverse folding task by further optimizing the architecture and decoding schemes. MIF-ST uses sequence transfer via a sequence masked language model to improve MIF. Similarly, Strokach et al. (2020) augments experimental structures by assuming that sequence homologs will fold to the same structure, while Hsu et al. (2022) augments GVP with 12 million structures predicted by AlphaFold2. In contrast to these methods, our focus is on learning

sequence from structure as a pretraining task instead of as the primary design task, and we therefore train a bidirectional denoising model instead of an autoregressive language model.

**Representations of protein structure** While we condition on structure and reconstruct sequence, there are other methods for incorporating protein structural information, such as predicting structure similarity between protein sequences (Bepler & Berger, 2019), corrupting and reconstructing the structure in addition to the sequence (Mansoor et al., 2021; Chen et al., 2022), encoding surface features (Townshend et al., 2019), contrastive learning (Zhang et al., 2022; Cao et al., 2021), or a graph encoder without sequence decoding (Somnath et al., 2021; Fuchs et al., 2020). LM-GVP uses the same architecture as MIF-ST consisting of a pretrained language model feeding into a GNN that encodes backbone structure (Wang et al., 2022). However, in LM-GVP the structure-aware module is used as a finetuned prediction head without any pretraining.

## 4 DOWNSTREAM TASKS

We evaluate MIF and MIF-ST on downstream tasks relevant to protein engineering, including out-of-domain generalization and zero-shot mutation effect prediction.

### 4.1 OUT-OF-DOMAIN GENERALIZATION

It is desirable for pretrained protein models to be able to make the types of out-of-domain predictions that often occur in protein engineering campaigns. For example, a protein engineer may want to train a model on single mutants and make predictions for sequences with multiple mutations, or train a model that is accurate for sequences with fitness greater than what is seen in the training set.

We finetune and evaluate on two fitness landscapes:

1. *Rma* NOD: Wu et al. (2019) explore how mutations at seven positions of the *Rhodothermus marinus* (*Rma*) nitric oxide dioxygenase (NOD) enzyme influences enantioselectivity on the reaction of phenyldimethyl silane with ethyl 2-diazopropanoate. We train on 214 variants with mutations at 4 positions and randomly split the variants with mutations at all 7 positions between 40 validation variants and 312 test variants. This tests the model's ability to generalize to mutations at unseen positions based on a small training set. We use PDB 6WK3 as the structure. Measurements were retrieved from ProtaBank (Wang et al., 2018).

2. GB1: Wu et al. (2016) performed a 4-site combinatorial deep mutational scan on protein G domain B1, an immunoglobulin-binding protein expressed in Streptococcal bacteria. We use splits from FLIP (Dallago et al., 2021) over the same GB1 landscape. These splits test generalization from fewer to more mutations or from lower-fitness sequences to higher-fitness sequences. We use PDB 2GI9 (Franks et al., 2006) as the structure.

We compare MIF and MIF-ST to CARP-640M, GVPMLM, and ESM-1b. All large models are finetuned end-to-end on a single Nvidia V100 GPU with a 2-layer perceptron as the predictive head until the validation performance stops improving. In addition, we use the small CNN from (Yang et al., 2022) and ridge regression as baselines.

As shown in Table 2, no model or pretraining scheme outperforms all others on both MSE and Spearman $\rho$ for the *Rma* NOD task. For protein engineering tasks, rank ordering is generally more important than minimizing error, so we will primarily compare the Spearmans. However, ridge regression achieves the best Spearman at the cost of a very high MSE. The small CNN is a strong baseline, with good performance by both metrics. MIF-ST outperforms CARP-640M and MIF with and without pretraining on Spearman. GVP generally does poorly on this task, while ESM-1b achieves the best MSE but is worse than MIF-ST and the small CNN on Spearman.

Table 3 shows results on the GB1 tasks. All models except GVP consistently benefit from pretraining on the GB1 tasks. Combining structure conditioning with sequence transfer seems to help slightly, with the biggest gains coming when the training set is limited to single- and double-mutants. However, for the most challenging 1-vs-many split, ridge regression results in the best performance, and for low-vs-high, the small CNN results in the best performance. MIF-ST with random weights consistently converged to a degenerate solution where it predicts the same value for all sequences in the test set for all 3 random seeds for 3 of the 4 splits.

Table 2: Out-of-domain prediction of *Rma* NOD enantioselectivity. Uncertainties are standard deviation over 3 random seeds.

| | MSE | | Spearman $\rho$ | |
|---|---|---|---|---|
| Model | pretrained | naive | pretrained | naive |
| CARP-640M | $0.14 \pm 0.03$ | $0.32 \pm 0.043$ | $0.69 \pm 0.05$ | $0.70 \pm 0.03$ |
| MIF | $0.12 \pm 0.05$ | $0.19 \pm 0.007$ | $0.66 \pm 0.11$ | $0.66 \pm 0.09$ |
| MIF-ST | $0.15 \pm 0.04$ | $0.19 \pm 0.004$ | $0.77 \pm 0.03$ | $0.73 \pm 0.05$ |
| GVPMIF | $0.18 \pm 0.004$ | $0.16 \pm 0.044$ | $0.27 \pm 0.67$ | $0.55 \pm 0.19$ |
| ESM-1b | $0.08 \pm 0.005$ | $0.18 \pm 0.001$ | $0.74 \pm 0.01$ | $0.69 \pm 0.05$ |
| small CNN | $0.12 \pm 0.02$ | | $0.76 \pm 0.009$ | |
| ridge | $1.17 \pm 0.01$ | | $0.80 \pm 0.005$ | |

In general, pretraining usually helps, as does adding structure when comparing MIF and MIF-ST to CARP-640M, and adding sequence transfer when comparing MIF-ST and CARP-640M. However, different tasks, even those using the same underlying protein fitness landscape, are not best predicted by the same models, and the baseline models compare favorably on many tasks. Furthermore, there is no correlation between pretraining performance and out-of-domain performance, even when adding structure or sequence transfer. This suggests a mismatch between the masked language model pretraining task and the sort of OOD performance desired for protein engineering.

Table 3: Performance on the FLIP GB1 tasks. Uncertainties are standard deviation over 3 random seeds. Values for CARP-640M, the small CNN, and ridge are taken from (Yang et al., 2022).

| | | Spearman $\rho$ | | | |
|---|---|---|---|---|---|
| | Model | 1-vs-many | 2-vs-many | 3-vs-many | low-vs-high |
| pretrained | CARP-640M | $0.19 \pm 0.26$ | $0.73 \pm 0.03$ | $0.87 \pm 0.004$ | $0.43 \pm 0.04$ |
| | MIF | $0.10 \pm 0.20$ | $0.71 \pm 0.02$ | $0.88 \pm 0.004$ | $0.38 \pm 0.06$ |
| | MIF-ST | $0.22 \pm 0.03$ | $0.74 \pm 0.03$ | $0.88 \pm 0.01$ | $0.43 \pm 0.01$ |
| | ESM-1b | $0.11 \pm 0.11$ | $0.67 \pm 0.07$ | $0.66 \pm 0.18$ | $0.42 \pm 0.09$ |
| | GVPMIF | $0.005 \pm 0.06$ | $0.66 \pm 0.04$ | $0.87 \pm 0.01$ | $0.44 \pm 0.07$ |
| naive | CARP-640M | $0.11 \pm 0.07$ | $0.38 \pm 0.26$ | $0.68 \pm 0.33$ | $0.23 \pm 0.26$ |
| | MIF | $0.03 \pm 0.11$ | $0.05 \pm 0.12$ | $0.23 \pm 0.02$ | $0.17 \pm 0.12$ |
| | MIF-ST | NaN | NaN | NaN | $0.18$ |
| | ESM-1b | $0.05 \pm 0.28$ | $0.14 \pm 0.13$ | $0.10 \pm 0.13$ | $-0.04 \pm 0.09$ |
| | GVPMIF | $0.17 \pm 0.09$ | $0.45 \pm 0.03$ | $0.83 \pm 0.01$ | $0.25 \pm 0.10$ |
| baselines | small CNN | $0.15 \pm 0.09$ | $0.39 \pm 0.04$ | $0.81 \pm 0.004$ | $0.47 \pm 0.01$ |
| | ridge | $0.28$ | $0.59$ | $0.76$ | $0.34$ |

## 4.2 ZERO-SHOT MUTATION EFFECT PREDICTION

Large language models can also predict experimental measurements of protein function without further training on sequence-fitness measurements or sets of evolutionarily-related sequences Hie et al. (2022b); Meier et al. (2021). Table 4 reports results on five datasets, which are described in A.1:

We score sequences by masking every mutated position and computing the log odds ratio between the mutated and wild-type residues at each mutated position, assuming an additive model when a sequence contains multiple mutations (the "masked marginal" method from Meier et al. (2021)) except for stability, where we use the pseudolikelihood. Where possible, we compare to ESM-1v, which is a transformer masked language model trained on UniRef90, Structured Transformer, the SE(3)-equivariant model from McPartlon et al. (2022), GVP, and GVP-AF2. The ESM-1x values for DeepSequence are taken from Meier et al. (2021); the ESM-1x values for RBD are taken from Hsu et al. (2022). We compute values for ESM-1v on MSP, stability, and GB1 using only the second

Table 4: Zero-shot effect prediction. We report average Spearman correlation for DeepSequence, Spearman correlation for RBD and GB1, AUROC for MSP, and Pearson correlation for stability.

| | | Task | | | | |
|---|---|---|---|---|---|---|
| Regime | Model | DeepSequence | MSP | RBD | stability | GB1 |
| sequence | CARP-640M | 0.493 | 0.53 | -0.05 | 0.28 | -0.08 |
| | ESM-1b | 0.457 | 0.55 | 0.02 | 0.26 | -0.04 |
| | ESM-1v | 0.508 | 0.54 | 0.03 | 0.22 | -0.05 |
| +structure | Structured Transformer | - | - | - | 0.37 | - |
| | GVP | - | 0.71 | 0.60 | 0.42 | - |
| | GVP+AF2 | - | 0.71 | 0.69 | 0.48 | - |
| | McPartlon et al. (2022) | - | - | - | 0.50 | - |
| | MIF | 0.402 | 0.71 | 0.59 | 0.45 | 0.24 |
| | MIF-ST | 0.509 | 0.64 | 0.55 | 0.47 | 0.23 |

model, not the full ensemble of five independent models. Values for GVP and GVP-AF2 are both taken from Hsu et al. (2022); we take the best reported value for each task across several tested model variants. Note that this GVP is trained on a different dataset and task than our GVPMLM model.

For all tasks except DeepSequence, MIF is better than sequence-only methods, and on DeepSequence, adding sequence transfer improves performance above that of the sequence-only methods. Within DeepSequence, MIF-ST beats CARP-640M on 22 out of 41 datasets and MIF on 37 out of 41 datasets. Figure A1 shows results for each of the DeepSequence datasets. On the other tasks, MIF and MIF-ST achieve similar results, with sequence-transfer not consistently improving zero-shot performance despite improving pretraining performance. We suspect this is because fitness is unidentifiable from observational sequence data alone (Weinstein et al., 2022), and therefore improved density estimation does not necessarily lead to improved zero-shot fitness predictions. Table A2 shows that MIF outperforms CARP-640M on all ten folds tested in the stability dataset, and MIF-ST outperforms MIF on six out of ten folds. On both MSP and stability, MIF and MIF-ST are comparable to other inverse folding methods, but GVP+AF2 is the clear winner on RBD.

Table 5: AlphaFold structures vs PDB structures.

| Model | Spearman $\rho$ | | | |
|---|---|---|---|---|
| | MIF | | MIF-ST | |
| | PDB | AF2 | PDB | AF2 |
| DeepSequence | 0.358 | 0.407 | 0.488 | 0.509 |
| GB1 | 0.202 | 0.24 | 0.199 | 0.23 |
| RBD | 0.59 | 0.18 | 0.55 | 0.18 |

For DeepSequence, GB1, and RBD, we also compared the effect of using PDB or AlphaFold structures, as shown in Table 5. Surprisingly AlphaFold structures lead to better predictions for both GB1 and DeepSequence. (We were only able to find PDB structures for 38 of the 41 DeepSequence datasets, so the results in Table 5 differ slghtly from those in Table 4. The PDB structures used are listed in Table A1.) As shown in Figure 3a, the AlphaFold structure for GB1 is nearly identical to its PDB structure. It is unclear why AlphaFold structures lead to better zero-shot predictions in these cases. However, using an AlphaFold-multimer prediction for RBD greatly degrades performance. Upon examining the structures, this is not surprising, as AlphaFold places the RBD on the wrong side of ACE2, as shown in Figure 3b.

Table A3 shows zero-shot performance on the GB1 dataset separated by number of mutations using both PDB and AlphaFold structures. Without structural information, CARP-640M performs poorly for even single mutants, and no correlation at all for triple and quadruple mutants. Adding structure allows MIF and MIF-ST to make much better predictions at all mutation levels, but the accuracy nevertheless falls very quickly as the number of mutations increases.

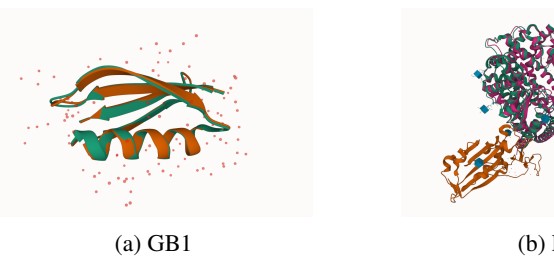

(a) GB1  (b) RBD

Figure 3: Comparisons of PDB (blue)and AlphaFold (red) structures for GB1 (PDB:2GI9) and SARS-Cov-2 RBD bound to human ACE2 (PDB:6M0J, orange & grey).

## 5  CONCLUSIONS

Protein structure is a richer source of information than protein sequence, but the largest protein sequence databases contain billions of sequences, while the number of experimental structures is currently in the hundreds of thousands. In this work, we investigate masked inverse folding on 19 thousand structures and sequences as a pretraining task. We observe that MIF is an effective pretrained model for a variety of downstream protein engineering tasks. We then extend MIF by transferring information from a model trained on tens of millions of protein sequences, improving pretraining perplexity and performance on some downstream tasks. High-quality predicted structures from AlphaFold often improve zero-shot performance over experimental structures. However, improving pretraining perplexity does not always lead to better downstream performance, and no model consistently outperforms the others on out-of-domain prediction tasks. We suspect that this is due to a mismatch between the masked language model pretraining task and out-of-domain fitness prediction

**Limitations** However, the MIF and MIF-ST pretraining schemes have several important limitations. First, they require structures as input during downstream tasks. This is ameliorated by the ability to predict high-quality structures for most protein sequences. In this work, we use a single structure for each protein and its variants: we may be able to improve results by predicting structures for all variants. However, this would be computationally expensive for large datasets, and it is currently unclear how good AlphaFold is at predicting the effects of single mutations (Pak et al., 2021). Some datasets, such as the the FLIP Meltome landscape, contain many unrelated sequences; collating or predicting structures for each sequence would be a significant endeavor. Furthermore, because the structure is held constant during pretraining, it is unclear how to deal with insertions and deletions in downstream tasks. For example, this prevented us from evaluating on the FLIP AAV landscape.

**Future work** The related work section suggests improvements that should be composable with MIF and MIF-ST. Using a more advanced GVP or SE(3)-transformer architecture instead of Structured GNN as the base model would likely improve pretraining performance, as would augmenting with AlphaFold structures or adding noise to the input structures. Another obvious extension is to train with an autoregressive or span-masking loss, which should be more amenable to generation tasks, better handle insertions and deletions, and may generalize better to complexes.

**Potential negative societal impacts** Machine learning on molecular data generally entails fewer societal risks than work on language, images, medical, or human data. Pretraining data comes from large, curated protein databases that compile results from the scientific literature, with no privacy concerns. However, large pretrained models incur significant energy and monetary costs to train. CARP-640M and ESM are trained on hundreds of V100s for weeks at a time, contributing to greenhouse gas emissions and putting retraining out of the reach of most academic labs.

**Outlook** Most work in protein pretraining has used methods borrowed from natural language processing on amino-acid sequences. However, leveraging information from structure, annotations, and even free text should improve performance. We hope that MIF and MIF-ST will lead to more investigations of multimodal protein pretraining tasks.

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

# A  ZERO-SHOT FITNESS PREDICTIONS

## A.1  DATASET

1. DeepSequence: 41 deep mutational scanning datasets originally compiled by Riesselman et al. (2018). These datasets each measure the effects of thousands of mutations or combinations of mutations to a parent sequence. Models are evaluated on their average Spearman $\rho$ across the datasets. We ensemble results from five AlphaFold structures obtained using ColabFold (Mirdita et al., 2021) default parameters for each wild-type protein.

2. Mutation Stability Prediction (MSP): Townshend et al. (2021) curate single mutants from SKEMPI (Jankauskaite et al., 2018), which measures whether mutant proteins display better binding than wildtype. Models are evaluated using area under the receiver operating curve (AUROC) on the 893 positive and 3255 negative examples in the test set.

3. Stability: Rocklin et al. (2017) used deep mutational scans to measure protease stability for point mutants of a set of *de novo* designed miniproteins with 10 different folds. Models are evaluated on their average Pearson correlation across the folds.

4. RBD: Starr et al. (2020) performed a deep mutational scan to measure how all amino-acid mutations to the receptor binding domain (RBD) of the SARS-CoV-2 spike protein affected its affinity to the human ACE2 receptor. Models are thus tasked to predict binding affinity for all 1311 mutant RBD sequences. We use PDB 6M0J (Rosas-Lemus et al., 2020) as the native structure.

5. GB1: Wu et al. (2016) performed a 4-site combinatorial deep mutational scan on protein G domain B1, an immunoglobulin-binding protein expressed in Streptococcal bacteria. Models are evaluated on Spearman $\rho$ across all measurements using an AlphaFold structure of the 56-amino acid GB1 domain.

## A.2  ADDITIONAL RESULTS

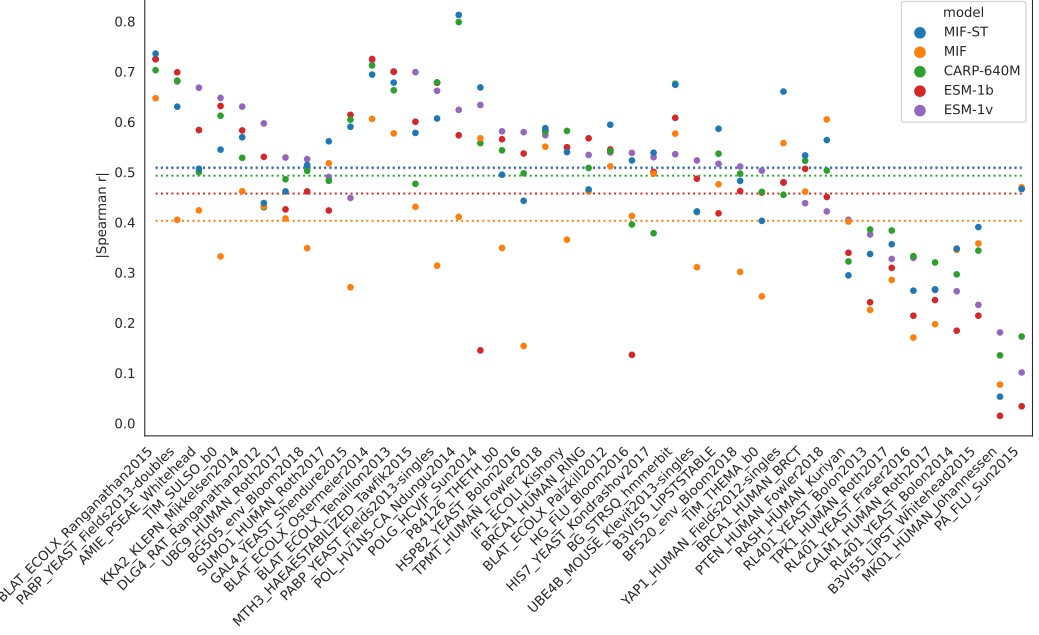

Figure A1: Zero-shot prediction on deep mutational scanning datasets in DeepSequence.

Table A1: Mappings from DeepSequence alignments to PDB structures.

| Alignment | PDB | chain_idx | aln_s | aln_e | pdb_s | pdb_e |
|---|---|---|---|---|---|---|
| AMIE_PSEAE_1_b0 | 2uxy | 0 | 0 | 341 | 0 | 341 |
| B3VI55_LIPSTSTABLE_1_b0 | 4zfv | 1 | 9 | 439 | 0 | 430 |
| B3VI55_LIPST_1_b0 | 4zfv | 1 | 9 | 439 | 0 | 430 |
| BG_STRSQ_1_b0 | 1gnx | 0 | 14 | 478 | 0 | 464 |
| BLAT_ECOLX_1_b0 | 1s0w | 0 | 0 | 263 | 0 | 263 |
| BRCA1_HUMAN_1_b0 | 1jm7 | 0 | 0 | 103 | 0 | 103 |
| BRCA1_HUMAN_BRCT_1_b0 | 4u4a | 0 | 21 | 235 | 0 | 214 |
| CALM1_HUMAN_1_b0 | 3sjq | 0 | 1 | 148 | 0 | 147 |
| DLG4_RAT_2_b0 | 2xkx | 0 | 0 | 101 | 296 | 397 |
| DYR_ECOLI_1_b0 | 5uio | 0 | 0 | 159 | 1 | 160 |
| F7YBW7_MESOW_1_b0 | 5ceg | 1 | 0 | 103 | 0 | 103 |
| FYN_HUMAN_1_b0 | 3uf4 | 0 | 5 | 66 | 0 | 61 |
| GAL4_YEAST_1_b0 | 3coq | 0 | 7 | 75 | 0 | 68 |
| HG_FLU_1_b0 | 6mya | 4 | 16 | 508 | 0 | 493 |
| HIS7_YEAST_1_b0 | 6ezm | 0 | 2 | 219 | 0 | 217 |
| HSP82_YEAST_1_b0 | 2cg9 | 0 | 1 | 216 | 0 | 215 |
| IF1_ECOLI_1_b0 | 1ah9 | 0 | 1 | 72 | 0 | 71 |
| KKA2_KLEPN_1_b0 | 1nd4 | 0 | 9 | 264 | 0 | 255 |
| MK01_HUMAN_1_b0 | 7opm | 0 | 0 | 360 | 2 | 362 |
| MTH3_HAEAESTABILIZED_1_b0 | 3ubt | 0 | 0 | 328 | 0 | 328 |
| P84126_THETH_1_b0 | 1vc4 | 0 | 0 | 254 | 0 | 254 |
| PABP_YEAST_1_b0 | 6r5k | 1 | 0 | 96 | 77 | 173 |
| PA_FLU_1_b0 | 7nj7 | 0 | 0 | 716 | 0 | 716 |
| POLG_HCVJF_1_b0 | 3fqq | 0 | 32 | 114 | 1 | 83 |
| POL_HV1N5-CA_1_b0 | 6wap | 0 | 0 | 231 | 0 | 231 |
| PTEN_HUMAN_1_b0 | 7jvx | 0 | 6 | 351 | 0 | 345 |
| PYP_HALHA_1_b0 | 4bbv | 0 | 0 | 125 | 0 | 125 |
| RASH_HUMAN_1_b0 | 4q21 | 0 | 0 | 169 | 0 | 169 |
| RL401_YEAST_1_b0 | 6zqh | 1 | 0 | 76 | 0 | 76 |
| SUMO1_HUMAN_1_b0 | 1a5r | 0 | 0 | 101 | 2 | 103 |
| TPK1_HUMAN_1_b0 | 3s4y | 0 | 15 | 242 | 0 | 227 |
| TPMT_HUMAN_1_b0 | 2h11 | 0 | 16 | 245 | 0 | 229 |
| TRPC_SULSO_1_b0 | 1igs | 0 | 1 | 248 | 0 | 247 |
| TRPC_THEMA_1_b0 | 1i4n | 0 | 1 | 252 | 0 | 251 |
| TRY2_RAT_1_b0 | 3fp6 | 0 | 0 | 223 | 0 | 233 |
| UBC9_HUMAN_1_b0 | 2xwu | 0 | 0 | 158 | 0 | 158 |
| UBE4B_MOUSE_1_b0 | 2kre | 0 | 8 | 104 | 4 | 100 |
| YAP1_HUMAN_1_b0 | 2ltw | 0 | 0 | 36 | 0 | 36 |

Table A2: Zero-shot performance on *de novo* miniproteins from Rocklin et al. (2017), broken out by fold.

| Fold | Pearson correlation | | |
|---|---|---|---|
| | CARP-640M | MIF | MIF-ST |
| $HHH_{138}$ | 0.41 | 0.46 | 0.52 |
| $HHH_{134}$ | 0.36 | 0.45 | 0.48 |
| $HEEH_{872}$ | 0.23 | 0.37 | 0.41 |
| $HEEH_{726}$ | 0.21 | 0.26 | 0.25 |
| $HEEH_{223}$ | 0.22 | 0.53 | 0.56 |
| $HEEH_{779}$ | 0.50 | 0.55 | 0.62 |
| $EEHEE_{1498}$ | 0.10 | 0.36 | 0.32 |
| $EEHEE_{37}$ | 0.41 | 0.65 | 0.68 |
| $EEHEE_{1716}$ | 0.22 | 0.60 | 0.60 |
| $EEHEE_{1702}$ | 0.03 | 0.26 | 0.24 |

Table A3: Zero-shot Spearman rank correlation on GB1 broken out by number of mutations.

| Model | PDB | | | | AF2 | | | |
|---|---|---|---|---|---|---|---|---|
| | 1 | 2 | 3 | 4 | 1 | 2 | 3 | 4 |
| CARP-640M | 0.19 | 0.12 | -0.01 | -0.01 | 0.19 | 0.12 | -0.01 | -0.01 |
| MIF | 0.83 | 0.64 | 0.42 | 0.11 | 0.84 | 0.66 | 0.44 | 0.15 |
| MIF-ST | 0.76 | 0.62 | 0.41 | 0.11 | 0.83 | 0.67 | 0.44 | 0.15 |

