# OpenReview forum: "Masked inverse folding with sequence transfer for protein representation learning"
_ICLR.cc/2023/Conference — Submitted to ICLR 2023_

### Official Review · Reviewer_mxA7 · 2022-10-14

**Confidence:** 4
**Correctness:** 3
**Technical Novelty And Significance:** 2
**Empirical Novelty And Significance:** 3
**Recommendation:** 6

**Clarity, Quality, Novelty And Reproducibility:**

Quality:

In general, this paper is well-written and well-motivated, except for some notation issues.


Novelty:

The technical novelty of this work is plain. The structure-based encoder is proposed by a previous work, and the masked inverse folding pre-training scheme share great similarity with the "structure-based residue type prediction" objective proposed by [a].


Reproducibility:

Authors submitted source codes for reproducing the results. The codes look well-organized according to my brief check.


**Strength And Weaknesses:**

Strength:
1. The proposed methods are well motivated. MIF is proposed to improve sequence-only masked language modeling with backbone structural information, and MIF-ST is proposed to further improve MIF by pre-trained protein sequence representations.
2. The advantages of pre-trained structure-based encoders are demonstrated on a comprehensive set of protein engineering related tasks, and the ablation study analyzes the effect of using experimental and predicted structures for downstream evaluation.


Weakness:
1. For performance comparison, some important protein structure pre-training baselines are lacked. In a closely related work [a], several structure-based encoders pre-trained by contrastive learning and self-prediction methods are proposed, and the source codes are released. It will be nice to compare with these baseline models on protein engineering tasks.
2. Some ablation studies are lacked. According to Table 2, ESM-1b and ESM-1v are superior protein language models against CARP-640M. Why not using their representations as inputs of MIF? In this way, we can expect stronger MIF-ST models than the current one based on CARP-640M representations.
3. Some notations are not consistent across figures and texts. In Figure 2, authors use $\varphi$ and $\theta$, while $\phi$ and $\psi$ are used in the equations (3) and (4).


[a] Zhang, Zuobai, et al. "Protein representation learning by geometric structure pretraining." arXiv preprint arXiv:2203.06125 (2022).


**Summary Of The Paper:**

This paper proposes to use masked inverse folding to pre-train (1) a protein structure-based encoder (MIF) and (2) a structure-based encoder enhanced by protein language model representations (MIF-ST). (1) MIF encodes the information of protein backbone structure and masked protein sequence into node and edge features, and the model is trained to recover the masked residues based on the encoded information. (2) MIF-ST further improves the performance of sequence recovery by feeding protein language model representations into the structure-based encoder. On zero-shot and out-of-domain protein landscape prediction benchmarks, MIF and MIF-ST achieve competitive empirical performance against previous methods.

**Summary Of The Review:**

To summarize, this paper shows some merits of boosting future research on structure-based pre-training for protein engineering, while more comprehensive performance comparisons and ablation studies are required to make it a more solid contribution. I think it is currently on the border and expect authors' efforts during the response period.

---

> ### Author Response · Authors · 2022-11-18
> **Response to reviewer mxA7**
>
> Thank you for your kind review. We have added additional performance comparisons and ablations. We will address specific comments below. We hope you will carefully consider the revision and rebuttal when deciding your final recommendations.
>
>
> >For performance comparison, some important protein structure pre-training baselines are lacked. In a closely related work [a], several structure-based encoders pre-trained by contrastive learning and self-prediction methods are proposed, and the source codes are released. It will be nice to compare with these baseline models on protein engineering tasks.
>
> Unfortunately, this data pipeline was too different from ours for us to run it during the discussion period. In their Supplemental material, they show very strong (SOTA) performance for some versions of their model on the GB1 low-vs-high task.
>
> >Some ablation studies are lacked. According to Table 2, ESM-1b and ESM-1v are superior protein language models against CARP-640M. Why not using their representations as inputs of MIF? In this way, we can expect stronger MIF-ST models than the current one based on CARP-640M representations.
>
> We agree that testing a different architecture could improve performance. However, the CARP-640M paper finds that CARP-640M outperforms ESM-1x on many OOD and zero-shot tasks, so we disagree that they are unequivocally superior for the tasks we consider. Unfortunately, it would be very expensive to train a large transformer on a dataset that doesn’t include anything with homology to our CATH test sequences. We believe we have mitigated this somewhat by adding results for ESM-1b and GVPMLM on the OOD tasks and ESM-1b and ESM-1v on the zero-shot tasks. In addition, we have added some simpler baseline models for the OOD tasks.
>
> >Some notations are not consistent across figures and texts. In Figure 2, authors use  and , while  and  are used in the equations (3) and (4).
>
> Good catch! We have fixed this.

---

> > ### Comment · Reviewer_mxA7 · 2022-11-28
> > **Feedback to Authors**
> >
> > Thanks for the efforts. In the revision, performance comparisons with more baselines are provided to demonstrate the proposed method (vs. ESM-1b and GVPMLM on OOD tasks; vs. ESM-1b and ESM-1v on zero-shot tasks). As suggested by other reviewers, the proposed pre-trained models do not achieve consistently superior performance on different benchmarks, which has not been addressed during rebuttal. Also, some pre-trained structure representation models are not compared with for the time limitation. I still admire the basic idea of combining protein sequence and structure information within a hybrid model, and think the paper is still on the border. I will not change my rating.

---

### Official Review · Reviewer_u96a · 2022-10-20

**Confidence:** 3
**Correctness:** 3
**Technical Novelty And Significance:** 2
**Empirical Novelty And Significance:** 2
**Recommendation:** 5

**Clarity, Quality, Novelty And Reproducibility:**

$Clarify\ and\ Quality$:

Some statements are either confusing or inappropriate. Some background knowledge is missing, e.g., dihedral and planar angles.

$Novelty$:

The idea of applying masked inverse folding is interesting. However, the technical novelty is limited as both Structure GNN and CARP have been proposed before. Overall, the novelty is okay to me because, in my opinion, performance matters more in pre-training methodologies.

$Reproducibility$:

The authors provided source code but the readme file is missing. Thus, it is unclear how to run the code.

**Strength And Weaknesses:**

$Strengths$:

Applying MLM to the protein inverse folding problem sounds interesting. It is valid to use graph neural networks to incorporate the structure priors from folded proteins. Besides, using the learned amino acid representations from a pretrained sequence-only protein MLM as inputs to the graph neural networks is a resonable and meaningful trial.

$Weaknesses$:

1. Performance.

Since this paper presents a straightforward pre-training methodology that uses masked inverse folding as a pretext task, it is important that the pre-trained model shows superior performance on downstream tasks. However, in Table 2, performance of the proposed MIF-ST is lower than GVP+AF2 on three selected tasks, i.e., MSP, RBD, and stability. Moreover, when compared to GVP only, MIF-SF still produces much worse results on MSP and RBD. Also, explanations behind these failures are missing.

Besides, when evaluated on the out-of-domain generalization tasks, CARP-640M is employed as the only baseline. However, MIF-ST already includes CARP-640M as the feature extractor for amino acids. Thus, the reported comparisons are kind of unfair because MIF-ST is expected to achieve better results than CARP-640M. The authors should at least include another recent baseline in Table 2. If there are no baselines available, the authors should clarify the reasons in the manuscript.

2. Some statements are either confusing or inappropriate.

In the abstract, the author claimed "inverse folding methods do not take advantage of sequences that do not have known structures." After reading the paper, I failed to find how the authors addressed this point besides incorporating the predicted structures from PDB and AF2 (as shown in Table 3). If this is the truth, the above statement should be revised because GVP already includes structures from AF2 as training data.

In the last paragraph of sec. Introduction, the sentence "Figure 1 compares our previous sequence-only dilated convolutional protein MLM (CARP), MIF, and MIF-ST" is not appropriate (from my perspective). The citation of CARP refers to the anonymous submission but the authors did not provide the anonymous draft. As a result, how to implement the CNN model in MIF-ST is missing (well, this is another problem that needs to be addressed in the rebuttal), and I have to search CARP online and find the draft on bioRxiv. I'm not sure if the use of words like "our previous xxx" violates author guidelines of ICLR'23 but personally, I recommend using other words instead.


Minor points:

1. Does formula 4 lack a concatenation notation? Otherwise, it should be $\mathcal{E}_{i,j}$.

2. In Figure 2, it seems that $w$ stands for the angle instead of coordinates.

3. Why do the authors take the best reported value instead of mean value for each task across several tested model variants?

4. Based on Table 3, I assume the authors already tried to use the predicted structures from AlphaFold2? If this is the truth, it is necessary to explain why GVP can utilize the predicted results from AF2 in a better way.

**Summary Of The Paper:**

For the first time, this study proposes to apply masked language modeling (MLM) to the inverse folding problem of proteins. The protein language model is parameterized as a bidirectional version of the structured GNN (Ingraham et al, 2019). The authors also showed that using the outputs from a pretrained sequence-only protein MLM as input can further improve pretraining perplexity. The proposed models are evaluated on downstream protein engineering tasks, including zero-shot mutation effect prediction and out-of-domain generalization.

**Summary Of The Review:**

I lean upon a weak reject due to the unsatisfactory performance compared to GVP on MSP, RBD, and stability. Moreover,

---

> ### Author Response · Authors · 2022-11-18
> **Response to u96a**
>
> Thank you for the kind and thorough review. We have added additional experiments and restructured the paper. We will address additional points below. We hope you will carefully consider the revision and rebuttal when deciding your final recommendations.
>
>
> >The authors should at least include another recent baseline in Table 2. If there are no baselines available, the authors should clarify the reasons in the manuscript.
>
> We have filled in the missing values for ESM-1b and ESM-1v in Table 2 (now Table 4).
>
> >In the abstract, the author claimed "inverse folding methods do not take advantage of sequences that do not have known structures." After reading the paper, I failed to find how the authors addressed this point besides incorporating the predicted structures from PDB and AF2 (as shown in Table 3). If this is the truth, the above statement should be revised because GVP already includes structures from AF2 as training data.
>
> We take advantage of sequences without known structures via sequence transfer. We have clarified this in the introduction: “We then show that using the outputs from a pretrained sequence-only protein MLM as input to \gnn\ further improves pretraining perplexity by leveraging information from sequences without experimental structures.”
>
>
> <Moreover, when compared to GVP only, MIF-SF still produces much worse results on MSP and RBD. Also, explanations behind these failures are missing.
>
> Because MIF-ST differs in architecture (GVP vs structured GNN), pretraining task (autoregressive span masking vs BERT), and dataset, it is difficult to determine why each model does better or worse on each task.
>
> |The authors provided source code but the readme file is missing. Thus, it is unclear how to run the code.
>
> There is already a public version of our code with nicely-packaged weights, scripts, and a README. Unfortunately, that version is integrated with many other tools and would be very difficult to anonymize.
>
> >Some statements are either confusing or inappropriate. Some background knowledge is missing, e.g., dihedral and planar angles.
>
> The angles are described in Figure 2. We are happy to provide additional background if you feel there are other specific missing pieces.
>
>
> > I'm not sure if the use of words like "our previous xxx" violates author guidelines of ICLR'23 but personally, I recommend using other words instead.
>
> This is a good point. Fixed.
>
> > Does formula 4 lack a concatenation notation? Otherwise, it should be Ei,j
>
> You are correct. It should be Ei,j. Thank you!
>
> >in Figure 2, it seems that
> >w
> >stands for the angle instead of coordinates.
>
> Yes, $\omega$ is an angle in Figure 2.
>
> >Why do the authors take the best reported value instead of mean value for each task across several tested model variants?
>
> The ESM-IF paper reports several variants of GVP and GVP+AF2. We take the best value of each to keep Table 4 easier to read and because it provides the toughest comparison against our models.
>
> > Based on Table 3, I assume the authors already tried to use the predicted structures from AlphaFold2? If this is the truth, it is necessary to explain why GVP can utilize the predicted results from AF2 in a better way.
>
> In Table 3, we use predicted structures at test time, whereas ESM-IF/GVP uses them for pretraining.

---

> > ### Comment · Reviewer_u96a · 2022-11-24
> > **Reply to Paper1987 Authors**
> >
> > Dear authors,
> >
> > After reading your reply, I found that most of my concerns were not addressed appropriately in your response. Considering some necessary details (of the methodology and experiments) are missing in the manuscript, I am afraid I have to remain my score unchanged.
> >
> > Best,

---

### Official Review · Reviewer_ABwW · 2022-10-23

**Confidence:** 4
**Correctness:** 4
**Technical Novelty And Significance:** 2
**Empirical Novelty And Significance:** 2
**Recommendation:** 5

**Clarity, Quality, Novelty And Reproducibility:**

Overall, the paper is well written and placed in the current research dialog. I found the structure of the body of the paper a bit confusing, though. There wasn't a clear separation between Methods and Results sections.

The proposed modeling idea combines a few core ideas already existing in the literature. It is a natural, well-motivated idea, but not ground-breaking.

**Strength And Weaknesses:**

=strengths=
Well written
Experiments compare to recent work
Experiments demonstrate that conditioning on structure can provide good performance improvements (e.g., on zero-shot modeling of the covid RBD).

=weakenesses=
The experiments don't clearly demonstrate a regime where the paper's proposed method improves over existing work. The experiments demonstrate that structure-conditioned models are good at zero-shot prediction, but existing structure-conditioned models can do this well too.

Experiments don't compare to any non-neural-network baselines

The paper is frank about the fact that there are lots of opportunities for follow-up work, such as by pretraining at scale on alphafold-predicted structures. The paper is good enough for acceptance in its current form, but it wouldn't have been a terrible idea to slow things down, run some of the proposed experiments, and submit when the full setup is available.


**Summary Of The Paper:**

BERT-style pretraining has become popular for protein modeling, since there are very large databases of natural protein sequences available. A separate body of work on inverse folding has explored structure-conditioned generative models of sequences.This paper explores structure-conditioned BERT, where at both train and test time the structure is assumed to be known for any query sequence. The authors demonstrate that conditioning on the structure improves performance on a variety of zero-shot and few-shot protein function prediction tasks.


**Summary Of The Review:**

I work in this exact research area and am in general supportive of this kind of work at ICLR. It can have large real-world impact and these protein modeling problems are also a great place for cutting-edge methods to be applied. My primary hesitation about this paper, however, is the nature of the experimental results.

I didn't understand why there were few results for ESM in Table 2. Without these, how can you make the claim "For all tasks except DeepSequence, MIF is better than sequence-only methods?" There is a good github package for ESM. It shouldn't be too hard to run on these datasets. Can you add these?

Along these lines, to me the take-away point from table 2 is that structure-conditioned models are good at zero-shot prediction, not that the particular model proposed by this paper provides a breakthrough. GVP, for example, is quite strong. Am I missing something? Where is there convincing evidence that MIF-ST improves over existing work?

I was disappointed by the complete lack of baselines that are not based on large neural networks. What about, for example, profile HMMs or Potts models for the zero-shot experiments? What about a simple linear model or lightweight non-linear model (like a shallow MLP) for the few-shot experiments? Adding all of these to the paper would be a lot of work. Can you provide baselines for at least some of the experiments, though?

The paper's propose model combines two orthogonal ideas: (1) using a CNN instead of a transformer and (2) conditioning on the structure. (1) appears in current work under review. It's not required for acceptance, but the paper would have been considerably improved if there were experiments that isolate (2) by using a transformer.

I was really surprised by how strong the RBD results are for structured-conditioned models. Do you have any intuitions for why conditioning on the structure makes such a big difference here?

---

> ### Author Response · Authors · 2022-11-18
> **Response to ABwW**
>
> Thank you for the thorough review. We especially appreciate this line:
>
> > The paper is good enough for acceptance in its current form.
>
> We have added additional experiments and analysis to clarify our contributions. We will address specific concerns below. We hope you will carefully consider the revision and rebuttal when deciding your final recommendations.
>
> ### Additional experiments
>
> > Experiments don't compare to any non-neural-network baselines
>
> > I was disappointed by the complete lack of baselines that are not based on large neural networks… What about a simple linear model or lightweight non-linear model (like a shallow MLP) for the few-shot experiments? Adding all of these to the paper would be a lot of work. Can you provide baselines for at least some of the experiments, though?
>
> We have added the following additional baselines on the OOD tasks:
> - GVP pretrained to perform masked inverse folding on CATH
> - ESM-1b
> - Ridge regression
> - A small CNN, as in FLIP
> We have also added additional text analyzing the results and the following synthesis:
>
> >In general, pretraining usually helps, as does adding structure when comparing MIF and MIF-ST to CARP-640M, and adding sequence transfer when comparing MIF-ST and CARP-640M. However, different tasks, even those using the same underlying protein fitness landscape, are not best predicted by the same models, and the baseline models compare favorably on many tasks. Furthermore, there is no correlation between pretraining performance and out-of-domain performance, even when adding structure or sequence transfer. This suggests a mismatch between the masked language model pretraining task and the sort of OOD performance desired for protein engineering.
>
> >What about, for example, profile HMMs or Potts models for the zero-shot experiments?
>
> Results for profile HMMs and Potts models are available in the original DeepSequence paper. However, we did not have time to compute them for the other zero-shot datasets (and neither does the ESM-IF paper).
>
> > I didn't understand why there were few results for ESM in Table 2. Without these, how can you make the claim "For all tasks except DeepSequence, MIF is better than sequence-only methods?" There is a good github package for ESM. It shouldn't be too hard to run on these datasets. Can you add these?
>
> We add the missing results for ESM-1b and ESM-1v in Table 2 (now Table 4). These results support our claim that “"For all tasks except DeepSequence, MIF is better than sequence-only methods.”
>
> >The paper's propose model combines two orthogonal ideas: (1) using a CNN instead of a transformer and (2) conditioning on the structure. (1) appears in current work under review. It's not required for acceptance, but the paper would have been considerably improved if there were experiments that isolate (2) by using a transformer.
>
> We agree! Unfortunately, it would be very expensive to train a large transformer on a dataset that doesn’t include anything with homology to our CATH test sequences. We believe we have mitigated this somewhat by including results for ESM-1b and GVPMLM on the OOD tasks.
>
>
> ### Other questions and concerns
>
> > I was really surprised by how strong the RBD results are for structured-conditioned models. Do you have any intuitions for why conditioning on the structure makes such a big difference here?
>
> Our intuition is that the structure-conditioned models have a principled way to access knowledge about the ACE2 receptor in addition to the RBD. Concatenating the ACE2 sequence to the RBD sequence does not improve performance when using CARP-640M (and results in sequences that are too long for ESM-1x).
>
> >I found the structure of the body of the paper a bit confusing, though. There wasn't a clear separation between Methods and Results sections.
>
> We went back and forth between the current format and a more traditional format that describes the pretraining and downstream modeling before presenting the results. We found that having each result closer to where the method for producing it is described made the paper easier to read. While this is not the most common structure, we note that the ESM-1b, ESM-1v, and ESM-IF papers are organized similarly.

---

### Official Review · Reviewer_vWP7 · 2022-10-24

**Confidence:** 5
**Correctness:** 4
**Technical Novelty And Significance:** 3
**Empirical Novelty And Significance:** 2
**Recommendation:** 5

**Clarity, Quality, Novelty And Reproducibility:**

Clarity: The paper is clear.

Quality: The quality is overall good but the experimeents in the OOD section could be improved.

Novelty: The novelty in the paper is not properly highlighted. The zero-shot section is not really novel while the experiments in the OOD section are a bit underbaked from a point that would help practitioners as well as methods development.

Originality: There is some originality but largely I suspect some of the core ideas of this work was scooped in recent months.

**Strength And Weaknesses:**

Strengths:
- Combining sequence and structural information and pretraining hasn't been done before which is definitely a natural progression for the progress in the protein + ML space in recent years.
- Finetuning of structural/inverse folding models hasn't been studied as much compared to sequence-based finetuning.

Weaknesses:
- The framing and experiments of the paper could be improved to highlight the core contributions. Hsu et al (2022) had demonstrated the impact of structure on zero-shot fitness prediction. The takeaway from the section on zero-shot prediction largely recapitulated the results from Hsu et al while demonstrating that adding sequence information atop structure did not seem to impact performance. While this is somewhat interesting, it seems like this should not be such a dominant section of the paper. Rather the pretraining and finetuning with structure and sequence for OOD prediction seems much more novel. However, the experiments in this section are much less explored and don't benchmark against simple baselines (CNNs etc) or other methods. Experiments focused on finetuning and pretraining strategies for structural data in combination with sequence data would be interesting but seemed less prioritized than the zero-shot section. Overall, the exposition and experiments seem to highlight the less novel application while providing less thorough experiments that would push the field forward for the OOD prediction task. Getting fine-tuning to work for these LLMs can be challenging so guidelines on how to do this for various representations(sequence-only, structure-only, combined) would be extremely valuable in my opinion.
- The point about the lack of structure data causing the need for sequence data for pretraining has been somewhat obviated given AlphaFoldDB.

Minor Comments:
- Citing ProteinMPNN (Daupras et al 2022) seems relevant here even if its not benchmarked against.

**Summary Of The Paper:**

The paper proposes using protein structure and sequence information as a pretraining task for representation learning of proteins. They then apply these methods to two downstream applications: (1) zero-shot prediction and (2) OOD prediction.


**Summary Of The Review:**

I think the paper has good work that pushes the research field forward for protein engineering applications. However, I think the focus on zero-shot prediction is mostly a distraction from the more useful aspects of combining sequence and structure information (as opposed to just structure and just sequence). More thorough experimentation and benchmarking along with a rework of the exposition to highlight this fact would greatly improve the quality of this paper.

---

> ### Author Response · Authors · 2022-11-18
> **vWP7**
>
> Thank you for your kind and thorough review. We especially appreciate your recognition of the novelty in our pretraining task and downstream analyses. We have performed additional experiments and reorganized the paper to better clarify and emphasize our contributions. We discuss specifics below. We hope you will carefully consider the revision and rebuttal when deciding your final recommendations.
> ### Additional experiments
> >However, the experiments in this section are much less explored and don't benchmark against simple baselines (CNNs etc) or other methods.
>
> We have added the following additional baselines on the OOD tasks:
> - GVP pretrained to perform masked inverse folding on CATH
> - ESM-1b
> - Ridge regression
> - A small CNN, as in FLIP
> - We have also added additional text analyzing the results and the following synthesis:
>
> >In general, pretraining usually helps, as does adding structure when comparing MIF and MIF-ST to CARP-640M, and adding sequence transfer when comparing MIF-ST and CARP-640M. However, different tasks, even those using the same underlying protein fitness landscape, are not best predicted by the same models, and the baseline models compare favorably on many tasks. Furthermore, there is no correlation between pretraining performance and out-of-domain performance, even when adding structure or sequence transfer. This suggests a mismatch between the masked language model pretraining task and the sort of OOD performance desired for protein engineering.
>
> ### Organization
> > While this is somewhat interesting, it seems like this should not be such a dominant section of the paper. Rather the pretraining and finetuning with structure and sequence for OOD prediction seems much more novel.
> > The novelty in the paper is not properly highlighted. The zero-shot section is not really novel while the experiments in the OOD section are a bit underbaked from a point that would help practitioners as well as methods development.
>
> This is a great point. We have restructured the paper to emphasize OOD prediction by adding additional baselines, experiments, and commentary as well as shortening the section on zero-shot predictions.
>
> ### Other concerns
>
> >Citing ProteinMPNN (Daupras et al 2022) seems relevant here even if its not benchmarked against.
> Thank you for pointing this out. We originally wrote this paper before ProteinMPNN came out. We have added it (and another more recent similar paper) to the related work.

---

> > ### Comment · Reviewer_vWP7 · 2022-11-28
> > **Response**
> >
> > Hi authors, I appreciate the effort in and additional experiments in the rebuttal phase -- I do sincerely believe there is potential within this work. However, understandably due to the time restriction the results do not justify the method nor are there careful ablation studies to provide practitioners with insights as to what could vs couldn't work, so my score remains unchanged. I'd encourage the authors if they seek to resubmit this work to focus on making this work either of two categories: (1) an interesting resource for the field as to what pretraining and finetuning strategies work well vs not in the OOD setting -- this would require careful experimentation and ablations much more than the single table provided (understandably due to the time constraints in the rebuttal phase) and/or (2) strong empirical performance that convinces the reader that this pretraining idea should be adopted.

---

### Author Response · Authors · 2022-11-07
**Summary author response**

We thank all reviewers for taking the time to review our paper. In general, we appreciate the reviewers' positive reception of the paper. The reviewers generally expressed that combining sequence and structure during pretraining and fine-tuning inverse folding models is a promising direction. Based on this, we are excited about the impact and discussion that our findings will generate in the protein machine learning community: as the reviewers suggest, our results point to many future experiments, such as using different sequence or structure models or using predicted structures.

We plan to clarify our claims and writing based upon the feedback from the reviewers. Several of the reviewers suggest that we emphasize OOD prediction over zero-shot learning, as this is the more novel application for our work. We agree, and we will restructure the paper to emphasize this. In addition, several reviewers request additional baselines. We propose to include the following additional baselines on the out-of-domain tasks:

- GVP-base pretrained to perform masked inverse folding on CATH
- ESM-IF
- ESM-1b
- Ridge regression
- A small CNN, as in [FLIP](https://www.biorxiv.org/content/10.1101/2021.11.09.467890v1)

We hope to have these results available in the next week. **We would appreciate any feedback from the reviewers on the proposed additional experiments in the meantime.**

 We will also address more specific responses to individual reviewers.

---

> ### Comment · Reviewer_vWP7 · 2022-11-11
> **My two cents on additional experiments**
>
> I generally think these choices of experiments are sensible. My main comment is that I would encourage the authors to strive for detailed ablation studies and analysis of pretraining and finetuning strategies to really strengthen the paper (similar to the Appendix of Meier et al 2021 from the ESM-1v paper). While this may be difficult to achieve in a week given the time constraints, I think it will be worth it as it will be a great contribution to the community.

---

> > ### Author Response · Authors · 2022-11-14
> > **Thanks!**
> >
> > Should have an update tomorrow or Wednesday with more OOD results. Let us know if you think of anything specific you might want to see!

---

### Author Response · Authors · 2022-11-18
**Summary of revisions**

Based on feedback from reviewers, we have performed extensive additional experiments and reorganized the paper. We believe the paper is much stronger as a result, so we encourage reviewers to take a look at the revision and the comments addressing their specific concerns when deciding their final scores. We would also encourage every reviewer to take to hear reviewer ABwW's claim that "The paper is good enough for acceptance in its current form" :-)

The most significant changes and additions follow:

### Additional experiments

We have added the following additional baselines on the OOD tasks:

- GVP pretrained to perform masked inverse folding on CATH
- ESM-1b
- Ridge regression
- A small CNN, as in FLIP

We did not end up using ESM-IF as a baseline because it requires a different structure preprocessing pipeline and only uses the sequence as input to some layers and not others. These changes meant we were not able to perform the experiments during the discussion period.

We have also added additional text analyzing the results and the following synthesis:

>In general, pretraining usually helps, as does adding structure when comparing MIF and MIF-ST to CARP-640M, and adding sequence transfer when comparing MIF-ST and CARP-640M. However, different tasks, even those using the same underlying protein fitness landscape, are not best predicted by the same models, and the baseline models compare favorably on many tasks. Furthermore, there is no correlation between pretraining performance and out-of-domain performance, even when adding structure or sequence transfer. This suggests a mismatch between the masked language model pretraining task and the sort of OOD performance desired for protein engineering.

In addition, we have filled in the missing values for ESM-1v and ESM-1b in the zero-shot results (formerly Table 2, now Table 4).

### Clarifying our contribution and conclusions

In order to emphasize our extensive experiments on the OOD tasks, we have moved that section up, expanded it, and shortened the zero-shot section.

We also clarify how we take advantage of "sequences that do not have known structures" in the introduction:

>We then show that using the outputs from a pretrained sequence-only protein MLM as input to MIF further improves pretraining perplexity by leveraging information from sequences without experimental structures.

---

### Decision · Program_Chairs · 2023-01-20

**Decision:**

Reject

**Justification For Why Not Higher Score:**

While the method is promising, all reviewers and AC agree that important experiments and comparison are lacking.

**Justification For Why Not Lower Score:**

N/A

**Metareview: Summary, Strengths And Weaknesses:**

The paper proposes two protein masked language models leveraging protein sequence and structure information for pre-training and subsequent fine-tuning on downstream tasks.

The reviewers and AC all agree that the paper makes some interesting contributions and the proposed methods are promising, but that further work is needed for the paper to be truly significant and impactful.

We strongly urge the authors to conduct additional experiments mentioned in the reviewer and discussion phase, which unfortunately could not be performed due to the time limitation.
- As noted by Reviewer vWP7, important ablation studies are missing and we strongly encourage the authors to perform them.
- It would also be important to compare against the additional pre-training baselines which could not be evaluated in time during the discussion phase.